# Brain Volumes and Cognition in Patients with Sickle Cell Anaemia: A Systematic Review and Meta-Analysis

**DOI:** 10.3390/children10081360

**Published:** 2023-08-08

**Authors:** Shifa Hamdule, Fenella J. Kirkham

**Affiliations:** 1Developmental Neurosciences Section, UCL Great Ormond Street Institute of Child Health, London WC1N 1EH, UK; shifa.hamdule.21@ucl.ac.uk; 2Clinical and Experimental Sciences, University of Southampton, Southampton SO16 6YD, UK

**Keywords:** sickle cell, brain volume, cognition

## Abstract

Cognitive decline is a major problem in paediatric and adult patients with sickle cell anaemia (SCA) and affects the quality of life. Multiple studies investigating the association between quantitative and qualitative neuroimaging findings and cognition have had mixed results. Hence, the aetiology of cognitive decline in this population is not clearly understood. Several studies have established cerebral atrophy in SCA children as well as adults, but the relationship between cognition and brain volumes remains unclear. The purpose of this systematic review was therefore to evaluate the literature on regional brain volumes and their association with cognitive outcomes. We also meta-analysed studies which compared regional brain volumes between patients and controls. Studies report that patients with SCA tend to have lower grey matter volumes, including total subcortical volumes in childhood as compared to controls, which stabilise in young adulthood and may be subjected to decline with age in older adulthood. White matter volumes remain stable in children but are subjected to reduced volumes in young adulthood. Age and haemoglobin are better predictors of cognitive outcomes as compared to regional brain volumes.

## 1. Introduction

While patients with sickle cell anaemia (SCA) are at risk of stroke and show persistent cerebrovascular damage [1], the effect of SCA pathology on brain volume reduction is not well defined. Earlier studies visually inspecting brain volumes showed some cortical atrophy [2,3]. With more recent advances in MRI techniques, the literature on brain volumes in this population is now increasing. However, there is still a poor understanding of how SCA pathology affects regional brain volumes, and the pathophysiological mechanisms underlying the reduction of brain volumes are still unclear.

Moreover, cognitive deficits which are commonly noted in patients with SCA, including a reduction in intelligence quotient (IQ), are well-documented in adult as well as paediatric patients with SCA [4], as well as in adults with anaemia in the general population [5]. Several studies have investigated various factors, such as the presence of silent cerebral infarcts (SCI) as well as transcranial Doppler velocities, to explain the cognitive deficits; however, very few studies have found a significant association. For example, studies looking at transcranial Doppler (TCD) velocities have found an association between mid-range to higher TCD velocities and short-term verbal recall [6]. In nine-month-old infants with SCA, higher TCD velocities and anaemia were associated with neurodevelopmental delay [7]. However, no association between TCD and IQ was found in older SCA children [8,9]. Investigations into MRI abnormalities have reported an association between IQ and SCI or lacunae, but the association diminished when age was added as a covariate [10]. Cognitive decline in sickle cell anaemia patients persisted even in the absence of SCI [4]. Another study at higher field strength found no association between lesion quantification and IQ [11].

Cerebral atrophy is a common finding in SCA populations [12,13]. However, the relationship between brain volume and cognition is not clearly established. There was an association between working memory index (WMI) and subcortical volume in adult SCA patients [14], while in our original paper [15], processing speed index (PSI) was associated with white matter volume (WMV) in adult and paediatric males with SCA. Other studies [16,17] on the relationship between brain volumes and cognition do not report similar findings.

MRI studies in typically developing children suggest associations with cognition. Grey matter density in regions such as the cingulate gyrus, orbitofrontal gyrus, cerebellum, and thalamus are associated with working memory, attention, and response selection [18]. Another study of brain volumes in typically developing children found an association between grey matter volume (GMV) in the anterior cingulate cortex and IQ in older children [19].

This suggests a case for associations between cognitive processes and regional brain volumes in patients with SCA. However, it remains to be established whether brain volumes are reduced in patients with SCA and if so, that explains the reduced cognitive scores in this patient population.

Hence, this systematic review aimed to objectively evaluate the literature on total and regional brain volumes in patients with SCA and their association with cognition. We also intended to meta-analyse regional brain volumes between patients with SCA and controls.

## 2. Materials and Methods

### 2.1. Search Strategy

Six databases (PubMed, Embase, Web of Science, Medline, Psychinfo, and Scopus) were used to conduct literature searches using terms including “brain volume” and “MRI” paired with “Sickle cell.” All studies from any time point until February 2023 were assessed based on rigorous inclusion/exclusion criteria (Table 1). Articles were eligible if they reported total or regional brain volumes based on MRI assessment. Case studies, editorials, conference abstracts, and reviews were excluded. References from excluded articles were searched for eligible studies. Authors were contacted for additional data. Studies in languages other than English were included only if they were translated into English.

### 2.2. Critical Appraisal

The Critical Appraisal Skills Programme (CASP) [20] Case-Control Checklist was used to assess the quality of the cross-sectional studies, while the CASP Cohort Study Checklist was used for longitudinal studies. Quality assessment was focused on the appropriateness of the control group (i.e., sibling/community controls or normative databases), the validity of the neuropsychological tools and the MRI methodology. All articles were graded on 11–12 questions with yes (1) or no (0) responses. Total scores were calculated, and articles were categorised as good (66% and above), satisfactory (36–65%), or poor quality (0–35%).

### 2.3. Meta-Analysis

Studies which compared the same regional brain volumes (grey matter volumes, white matter volume and subcortical volumes) between patients and controls were included in the meta-analysis. A random effect meta-analysis was performed to estimate a summary estimate of the standardised mean difference (Cohen’s d), and heterogeneity was assessed using the Q-test. The amount of variation related to the heterogeneity was represented using the I^2^ statistic. All statistical analyses were performed using the Comprehensive Meta-analyses Software V3.

### 2.4. PRISMA Statement

The Preferred Reporting Items for Systematic Reviews and Meta-Analyses (PRISMA) 27-item checklist was used to report the results of this systematic review.

## 3. Results

The literature search resulted in 451 articles. After removing 61 duplicates, 390 articles were screened based on titles and abstracts. Reasons for exclusion are mentioned in the Preferred Reporting Items for Systematic Reviews and Meta-Analyses (PRISMA) [21] flow chart (Figure 1). Thirty-five full-text articles were assessed for eligibility based on the inclusion and exclusion criteria. Twenty articles were retained for the systematic review, of which six were included in the meta-analysis.

### 3.1. Characteristics of the Study

Of the twenty articles that analysed brain volumes, thirteen articles (65%) were cross-sectional studies, four (20%) were prospective cohort studies, and three (15%) were longitudinal studies. Most studies (60%) had control groups. The sample size ranged from 25–312 patients and control groups ranged from 21 to 71 participants. The mean ages for the studies ranged from 8.4 to 34.3 years. Five studies performed analysis stratified by the presence of SCI. Publication years for the studies ranged from 1996–2023, of which two studies were published before 2000 and the rest of the studies were between 2000–2023. Sixteen studies were conducted in the USA, three studies were from the UK and one study was from Tanzania. Most studies analysed mean haemoglobin, age, and sex, as well as intracranial volumes as covariates, while some studies analysed socioeconomic status measured by education deciles and family income, as well as blood oxygen measures (SpO_2_).

Ten studies (50%) analysed brain volumes alongside cognitive outcomes, and one study analysed pain prevalence alongside grey matter volume (GMV). Most studies used the Weschler Scales for intelligence to measure cognitive function. Two studies used K-BITS and the Tanzanian study used Raven’s Progressive matrices to measure IQ. One study analysed executive functioning alongside intelligence quotient measurement using the Delis–Kaplan Executive Function Scales (D–KEFS) as well as the Test of Everyday Attention (TEA).

The magnetic field strength of MRI scanners ranged from 0.5 Tesla to 3.0 Tesla, except for one study that analysed hippocampal volumes at 7 Tesla. Various methods of analysing brain volumes were used. Studies (15%) before 2000 visually inspected and graded atrophy based on severity. Studies after 2000 have used various automated and semi-automated techniques and software to evaluate regional brain volumes. Four studies used various versions of SPM, of which two studies report the grey matter and white matter densities using Voxel-Based Morphometry (VBM) analysis. Two studies used FSL, four used FreeSurfer, two used SIENA, two used BrainSuite, one study used Photoshop and one study used Surgical Navigation Technology (See Appendix A). 

### 3.2. Critical Appraisal

The quality of the studies was assessed using two CASP checklists for cohort studies and case-control studies. Eighty-five per cent of the studies were graded good, while two studies were graded satisfactory. The results of the critical appraisal are summarised in Table 2.

### 3.3. Meta-Analysis

Eight studies out of twenty compared regional brain volumes between patients and controls. Of these, there was missing data for two studies which were excluded from the analyses. Three studies compared regional grey matter volumes between patients and controls, three studies compared white matter volumes (WMV) between patients and controls and five studies compared subcortical volumes between patients and controls. A separate random-effects meta-analysis was conducted for each ROI. The result of the meta-analysis is summarised in forest plots (Figure 2). There was a significant mean effect size for three studies comparing GMV between patients and controls of −0.597 ((95% CI = −0.861 to −0.332); Heterogeneity: Tau^2^ = 0.013, Q-value = 2.581, df(Q) = 2, I^2^ = 22.5, Z = −4.42). For the WMV, the standardised mean difference between patients with SCA and controls was non-significantly lower: −0.636 ((95% CI = −1.556 to 0.283); Heterogeneity: Tau^2^ = 0.618, Q-value = 32.17, df(Q) = 2, I^2^ = 93.7, Z = −1.35). Five studies analysed the mean difference for subcortical between patients and controls. The standardised mean difference was −1.478 ((95% CI = −2.966 to 0.012) Heterogeneity: Tau^2^ = 2.843, Q-value = 225.48, df(Q) = 4, I^2^ = 98.22, Z = −1.94), suggesting a trend for lower total subcortical volumes in patients.

### 3.4. Outcomes

A.Grey Matter Volumes

Out of twenty studies, ten studies analysed grey matter volumes, of which six studies analysed grey matter volume differences between patients with SCA (two studies stratified by SCI) and controls, and four studies analysed grey matter volumes only in patients with SCA. Studies comparing GMV in children with SCA and controls tended to report lower grey matter volumes in the frontal and parietal lobes [22,30]. GMV was further reduced in patients with SCA who tended to have SCI lesions and vasculopathy [1,27]. One longitudinal study analysed grey matter volume change over a period of four-time points. The authors of this study reported that grey matter volumes were reduced by 411 mm^3^ and were linearly associated with age in children with SCA, while GMV was reduced by 227 mm^3^/yr in controls and was quadratically associated with age [23]. The SIT trial data compared brain volume change longitudinally between transfused and patients with standard treatment; GMV was reduced by 0.9%/yr and transfusion status did not affect the reduction in brain volume percentage [13,28]. Two studies in adults did not report any differences in GMV [16,25].

Four studies (40%) assessed the association between GMV and IQ; the results are inconsistent, attributable to differences in methodologies. However, all studies make a meaningful contribution to the existing literature. Two studies out of four did not recruit any control groups and used K-BITS to measure IQ [22,24]. The cross-sectional study by Chen et al. (2009) used GAMMA to investigate the regions of GMV that correlated strongly with IQ. They found decreased GMV in the frontal lobes, including the frontal medial orbital gyrus, the superior frontal gyrus, the parietal lobe, including the supramarginal and angular gyri, and the temporal lobe, including the parahippocampal gyrus, superior, middle, and inferior temporal gyrus, and fusiform gyrus, areas associated with lower IQ in neurologically intact children [22]. The longitudinal observational study examined the effect of volumes and cognition in the same cohort over a period of 5 years and divided the patients into decline and non-decline groups. Patients in the decline group tended to have higher IQ at baseline, but also lower GM volumes in five out of six regions associated with K-BITS decline. These children also had a higher incidence of SCI, which predicted a K-BITS decline over the 5 years [24]. However, these results had low generalisability as the sample size was low in both studies. Other studies looking at this association were from unpublished datasets. The analysis did not reveal any associations between GMV and IQ measured on Weschler Scales for Intelligence in paediatric or adult SCA patients [15], nor were any associations seen with Raven’s Progressive Matrices in Tanzanian children with SCA [27]. Although IQ remained lower in patients across all studies, GMV was significantly lower only in Tanzanian SCA children [27].

B.White Matter Volumes

Eleven studies out of twenty analysed WMV. Seven studies compared WMV between patients and controls, of which one was stratified by SCI [1]. Results for reduced WMV were contradictory. Four studies reported reduced WMV, while three studies did not report any reductions. Patients with SCA and SCI tended to have reduced white matter densities in regions along the MCA territories [1]. One study in young adults found a reduced WMV of 8.1% in the right hemisphere and 6.8% in the left hemisphere [26]. In studies that did not report reduced WMV in patients, only one article reported results in paediatric patients [31]. In a longitudinal study of children with SCA, WMV increased at a lower rate as compared to controls. WMV was linearly associated with age in patients with SCA but was quadratically associated with age in controls [23].

Only one published study found that WMV predicted IQ in male adolescents and young adults with SCA [26]. In this study, tensor-based morphometry (TBM) was used to create a mean deformation index for WMV, which was higher in male patients, as was the burden of SCI, and positively correlated with IQ and haemoglobin in SCA patients. Additionally, lower WMV correlated positively with anaemia severity in the bilateral frontal, temporal and parietal lobes. Two other studies revealed no association between WMV and IQ in SCA patients or controls. WMV, however, was associated with PSI only in male patients in one of the studies [15]. Haemoglobin appeared to be a predictor of WMV and IQ in both male and female patients [26], suggesting that early exposure to anaemia with compensatory cerebral haemodynamic mechanisms influences WMV. Oxygen is carried through haemoglobin in red blood cells. When haemoglobin is lower, cerebral blood flow (CBF) increases to compensate for lower oxygen, also increasing the risk of stroke. While GMV in young adults is relatively preserved, the white matter may remain hypoxic, resulting in lower volumes and an increased risk for microstructural damage [15,26].

C.Subcortical Volumes

Six studies examined total subcortical volumes, while one study evaluated hippocampal subfields at 7T in patients with SCA and controls [29]. Subcortical volume reduction was noted only in 2/5 studies in SCA children and was not associated with cognitive outcomes in any of the five. Both studies noted significant reductions in the hippocampus, amygdala, and pallidum bilaterally, while some regions, such as the right thalamus and accumbens, were spared [17,27]. Moreover, SCA patients with SCI appeared to have lower subcortical volumes [17,27].

Adult studies have had mixed results in subcortical volumetric reductions. Two studies did not show any reductions in subcortical volume [15] and total hippocampal volume [16], nor were they associated with IQ, performance IQ [16] or working memory [15]. All four studies noted anaemia severity as a strong predictor of neurocognitive scores as well as volumetric reduction in SCA. Mackin et al. [14] found that adult SCA patients tended to have lower basal ganglia and subcortical volumes. They also showed an association with reduced WMI in SCA patients. Contrary to other studies, haemoglobin was not associated with cognitive outcomes in SCA patients or controls in this study. The authors noted that subcortical volumes and cognition may be affected by inflammation and other disease-related pathologies such as multiorgan dysfunction, sleep apnoea, arthritis, and chronic pain [14]. Age significantly predicted volumetric reduction and cognitive decline in SCA patients [15,16,27]. Hippocampal volume is particularly vulnerable to atrophy with age, which may be associated with cognitive decline in older adults [14].

D.Total Cortical Atrophy

Two studies before 2000 reported total cortical atrophy in patients with SCA [2,3]. These studies used visual inspection of atrophy on MRI scans and presented the prevalence of atrophy in patients with SCA. All studies report data on paediatric patients with SCA. One study [2] reported MRI-related outcomes in 312 patients, of which 15 patients showed only atrophy while 20 showed atrophy and infarction. Focal atrophy was noted in 5/9 patients and was seen in the frontal, occipital, and temporal lobes. They also noted that atrophy was mainly seen in patients over the age of 30 years [2]. Similarly, another study noted generalised atrophy in 2/146 patients aged 12 years above [3]. Moser et al. compared patients by genotype status and found that 8/25 patients that showed atrophy were of SS genotype.

## 4. Discussion

In this article, we reviewed the literature on regional brain volumes in patients with SCA. Studies had differing objectives and various scanner types and MRI processing techniques, making deriving robust conclusions challenging. However, certain underlying trends do exist in the literature. Consideration of the dynamic nature of brain development is of utmost importance in understanding the implications of SCA disease pathology on regional brain volumes. Brain volume development tends to be delayed in patients with SCA, which may be a result of abnormal cerebral haemodynamics at an early age. Hence, regional brain volumes, specifically GMV, tend to be lower in paediatric patients, stabilise during young adulthood, and may be vulnerable to increased damage during adulthood. More than volumes, haemoglobin and oxygen delivery are influential in cognitive functioning.

### 4.1. Grey Matter Volumes

Out of six papers that compared GMV between patients and controls, four studies showed reduced volumes in patients with SCA. Three of the six studies involved paediatric patients, which all noted reduced GMV. Another longitudinal study in children with SCA also noted a linear reduction of GMV with age in paediatric patients with SCA as compared to their typically developing peers, who show a more stabilised quadratic relationship between GMV change and age. In contrast to the reduced volumes in young children and adolescents [1,27,31], studies in adults with SCA did not show reduced GMV [15,16,25]. While this may seem like a discrepancy in the literature on the surface, this “catch-up” of GMV could be attributed to delays in the early development of GMV. In typically developing children between the ages of 3–15 years, GMV tends to reduce with age due to developmental processes such as synaptic formations as well as synaptic pruning [33]. These developmental processes start early (around the age of 3), and increasing myelination could result in steady declines of GMV until the ages of 9–11 years [31,33]. In young children with SCA, this initial process of synaptic formation and pruning is likely delayed, resulting in a delay of GMV decline beyond the age of 9 years which seems accelerated when compared to their TD peers in late childhood (9 years) [23,31]. Both studies that recruited young adults did not have patients below the age of 12 years, which makes this hypothesis plausible [16,25]. However, more longitudinal studies in larger samples and wider age groups need to be conducted to investigate GMV change in patients with SCA.

Chen et al. [22] found an association between grey matter volumes and IQ variability in low-IQ SCA patients, while other studies did not report any associations with cognition. This discrepancy with other studies could be explained by one of the recruitment factors. Chen et al. (2009) observed this relation only in neurologically intact SCA children with below-average IQ, which is a small subset of SCA patients, reducing the generalisability of the results. Other studies comparing the association of GM volumes with IQ did not find any association between the two [15,27]. However, Chen et al’s contribution towards a prognostic model of IQ decline in SCA children is commendable [24].

### 4.2. White Matter Volumes

Most studies in this systematic review noted reduced WMV in patients with SCA. One of the studies only found reduced WMV in male patients [26]. Choi et al. (2019), in their study, noted that anaemia severity was associated with lower WMV in watershed areas of the brain. WM density has also been reported as lower in SCA patients compared with controls [1]. Haemoglobin levels are a marker of oxygen delivery to the brain. In anaemia, cerebral blood flow increases to compensate for lower oxygen delivery [34]. Increased CBF also increases the risk of white matter structural damage mainly seen in watershed areas as the ceiling for further CBF increase in response to increased metabolic demand is exceeded [11]. Males tend to have larger brain volumes compared to females [31]. Hormonal factors, such as high oestrogen levels, may have a protective mechanism in females, leaving males more vulnerable to SCA disease severity, potentially explaining reduced WMV in males with SCA [26].

WMV was associated with IQ only in one study, while no other studies reported any association between WMV and cognition [26]. Likely, WMV is not very influential in cognitive processes, but the microstructural integrity of white matter tracts could influence processing speed underlying other cognitive output [35]. White matter tracts, studied using DTI, are particularly vulnerable to hypoxic damage, especially along the watershed areas [11], and are associated with processing speed which may contribute to poor cognition in patients with SCA.

### 4.3. Subcortical Volumes

All studies in paediatric populations tend to show reduced subcortical volumes in the patient group. Studies with adults show mixed findings. One study showed no reductions in total subcortical volumes in patients and controls with SCA, while another study that looked at basal ganglia and thalamus reported reduced volumes in the patient group. Two studies looked at hippocampal volumes, while one of them investigated 7T MRI data. Both these studies report reduced hippocampal volumes. Similar developmental trends that are attributable to GMV development may explain the discrepancy in the literature. While the developmental delay may explain reduced subcortical volumes in paediatric patients with SCA [31], accelerated ageing and poor cerebrovascular mechanisms may explain reduced volumes in older patients with SCA [14,29].

Only one study in adults showed an association between reduced basal ganglia volumes and reduced working memory and performance IQ in general, which could be attributed to age-related effects [14].

### 4.4. Influence of Cerebral Haemodynamic

Most studies supported the role of haemoglobin in predicting neurocognitive outcomes. In SCA patients as young as nine months old, anaemia predicted neurodevelopmental delay measured on Bayley’s Infant Neurodevelopmental Scale [7]. Haemoglobin levels were also associated with short-term verbal memory in SCA children [36]. Vichinsky et al. [16] also found that the severity of anaemia was associated with age-related decline in SCA adults. Studies of oxygen extraction fraction and CBF have shown a negative association with processing speed index and working memory [34]. Steen et al. [31] also found that 27% of the variability in IQ was related to haemoglobin levels in SCA children. All this evidence suggests that compared to brain volumes, haemoglobin levels are a better predictor of IQ.

### 4.5. Limitations

Most studies in this review were conducted on the same cohorts in high-income countries. It is noted that brain development, as well as cognitive trajectories, in low- and middle-income countries, may be notably different from high-income countries, influenced by factors such as nutrition, maternal education, socioeconomic status, and sex [37]. Only one paper in this review looked at research from a low-middle-income country, limiting the generalisability of results. This review also considered evidence from the author’s original paper, which may introduce bias. Studies in this review used varying neuropsychological tests and MR data processing steps, making direct comparison challenging.

## 5. Conclusions

In summary, cognitive difficulties in SCA persist regardless of SCI presence [4]. Lower GM volumes may be associated with IQ decline in neurologically intact SCA children [22,24]. This suggests that brain morphology may be a marker of cognitive decline in neurologically intact SCA children. However, further analysis with bigger sample sizes must be considered to increase the generalisability of the findings. Haemoglobin is the best predictor of IQ in SCA. Efforts to measure haemoglobin and targeted interventions to keep steady haemoglobin levels would be effective strategies to reduce cognitive decline. Male SCA patients may be more vulnerable to severe disease as compared to female patients [26]; increasing haemoglobin should be considered as a part of the treatment plan [34]. Age-related cerebral atrophy is a marker of SCD pathology in both paediatric and adult patients [28], but it is unclear if it explains cognitive difficulties and decline. Further research can explore the relationship between cerebral atrophy and cognitive decline as a function of age.

## Figures and Tables

**Figure 1 children-10-01360-f001:**
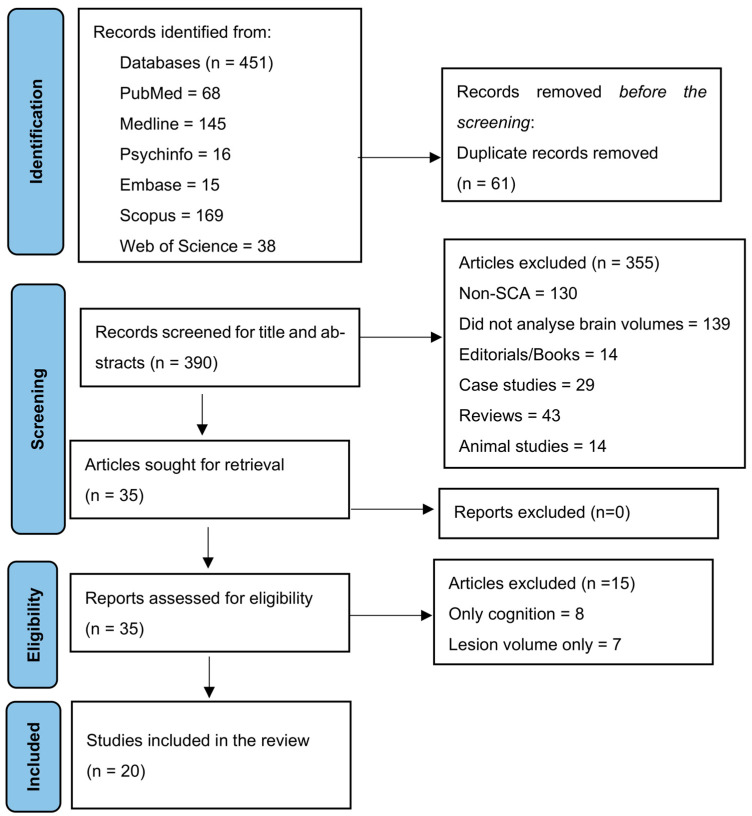
PRISMA flowchart.

**Figure 2 children-10-01360-f002:**
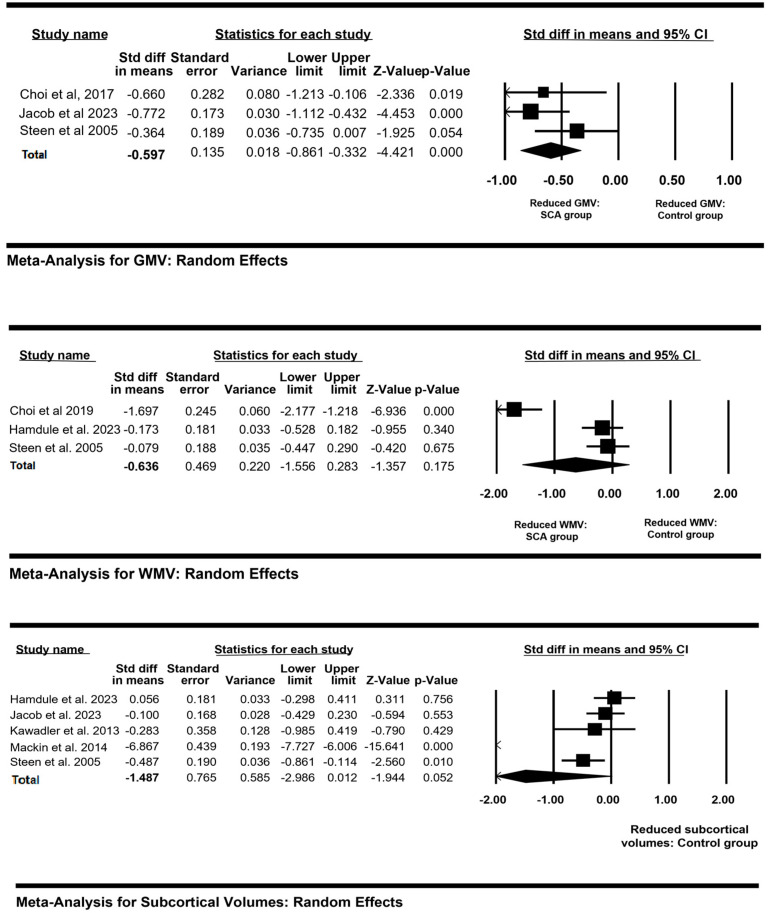
Forest plots for regional brain volumes between patients with SCA and controls. References: [14,15,17,25,26,27,31].

**Table 1 children-10-01360-t001:** Inclusion/exclusion criteria.

Inclusion Criteria	Exclusion Criteria
*Types of studies*: Participants of all ages; Studies published between any time until February 2023; All SCD genotypes; Original studies only; Studies in other languages (translated) or countries.	*Types of studies*: Systematic reviews or other reviews; No original data; Case studies; Editorials.
*Methodological aspects*: Studies investigated brain volumes on sMRI; Used standardised cognitive battery to assess cognition in case of studies including cognition; All study designs.	*Methodological aspects*: Only cognition fMRI, DTI, or surface-based analysis.

Note: Abbreviations: MRI = magnetic resonance imaging; sMRI = Structural MRI; fMRI = functional MRI; DTI = Diffusor Tensor Imaging.

**Table 2 children-10-01360-t002:** Quality Appraisal.

Author and Year	Q1	Q2	Q3	Q4	Q5	Q6	Q7	Q8	Q9	Q10	Q11	Q12	Grade
Baldweg et al., 2006 [1]	1	1	1	1	1	1	1	1	1	1	1	-	Good
Chen et al., 2009 [22]	1	1	0	0	1	1	1	1	1	0	0	-	Satisfactory
Chen et al., 2015 [23]	1	1	1	1	0	1	1	1	1	1	1	1	Good
Chen et al., 2017 [24]	1	1	1	1	0	1	1	1	1	1	1	1	Good
Choi et al., 2017 [25]	1	1	1	1	1	1	1	1	1	1	1	-	Good
Choi et al., 2019 [26]	1	1	1	1	1	1	1	1	1	1	1	-	Good
Hamdule et al., 2023 [15]	1	1	1	1	1	0	1	1	1	1	1	-	Good
Jacob et al., 2023 [27]	1	1	1	1	1	1	1	1	1	1	1	-	Good
Kawadler et al., 2013 [17]	1	1	1	1	1	1	1	1	1	1	1	1	Good
Kawadler et al., 2017 [28]	1	1	1	1	1	1	1	1	1	1	1	-	Good
Mackin et al., 2014 [14]	1	1	1	0	1	1	1	1	1	0	1	-	Good
Manfre et al., 1999 [2]	1	1	1	0	1	1	0	1	1	0	1	-	Satisfactory
Moser et al., 1996 [3]	1	1	1	1	1	1	0	1	1	1	1	-	Good
Santini et al., 2021 [29]	1	1	1	1	1	1	1	1	1	1	1	-	Good
Steen et al., 2003 [30]	1	1	1	0	1	1	0	1	1	1	1	-	Good
Steen et al., 2005 [31]	1	1	1	1	1	1	1	1	1	1	1	-	Good
Vichinsky et al., 2010 [16]	1	1	1	1	1	1	1	1	1	0	1	-	Good
Wang et al., 2022 [32]	1	1	1	0	1	1	1	1	1	1	1	1	Good
Chen et al., 2017 [24]	1	1	1	1	1	0	1	1	1	1	1	1	Good
Darbari et al., 2014, 2018 [13]	1	1	1	1	1	0	1	1	0	1	1	1	Good

Note: 1—yes; 0—no; good—66% and above; satisfactory—36–65%; or poor quality—0–35%.

## Data Availability

Data will be made available on a direct request from the authors.

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
