# Peer review of "Brain Volumes and Cognition in Patients with Sickle Cell Anaemia: A Systematic Review and Meta-Analysis"

_children, 2023, doi:10.3390/children10081360_

Round 1
Reviewer 1 Report
Dear authors, great effort by you guys. Please find some minor suggestions in the attached PDF

Grammatically sound
Author Response
We thank you for taking the time to go through our manuscript. We have attached the responses in the Word file below. Please see the attachment.
Thanks,
Shifa H.

Reviewer 2 Report
The methodology is sound in most respects. However, there is no statement regarding prior registration. Failure to include reports on languages other than English, unless a translation was available, is a weakness, especially as SCA is prevalent in African countries, not all of which use English as the official language. As the authors acknowledge, there is only one study from an African country.
The PRISMA chart has unexplained features, Only 283 of the 355 reports excluded at screening are accounted for. Furthermore, it is not clear where the 2 articles discovered from author correspondence were added as 35 were sought for retrieval, of which 15 were excluded, yet the final total is 20.
The use of the CASP checklist to assess quality is a positive feature. The quality of 17 of the included studies is good, and 2 are satisfactory. However, there is no quality assessment reported for one of the 20 included studies.
In Figure 2, the third panel depicting forest plot for the overall effect in the analysis of subcortical volume does not match the effect reported in the text and does not appear plausible. This apparently reflects an implausibly large z value for the effect reported by Mackin et al 2014.
The statement regarding hippocampal atrophy in line 230-231 is incorrectly attributed to Mackin et al 2014.
Several sets of initials are employed without definition (SCI, PSI, SCD).
In light of the fact that the references are numbered according to Vancouver style, it would be helpful if the reference number for each study was given in table 3.
Minor issue: The word ‘Scales’ is mis-spelled in line 109.
Author Response
We thank you for your consideration of our manuscript. We have provided the responses in the Word file below. Please see the attachment.
Regards,
Shifa H.
